# Low Galectin-3 Expression Level in Primary Tumors Is Associated with Metastasis in T1 Lung Adenocarcinoma

**DOI:** 10.3390/jcm9061990

**Published:** 2020-06-25

**Authors:** Ming-Wei Kao, Yue-Chiu Su, Peir-In Liang, Yi-Ying Wu, Tse-Ming Hong

**Affiliations:** 1Division of Thoracic Surgery, Department of Surgery, E-Da Hospital, and College of Medicine, I-Shou University, Kaohsiung 82445, Taiwan; kao.m.wei@gmail.com; 2Institute of Clinical Medicine, College of Medicine, National Cheng Kung University, Tainan 70101, Taiwan; yiying510@gmail.com; 3Department of Pathology, Kaohsiung Medical University Hospital, Kaohsiung Medical University, Kaohsiung 80756, Taiwan; syc0804@yahoo.com.tw (Y.-C.S.); peirinl@yahoo.com (P.-I.L.); 4Clinical Medicine Research Center, National Cheng Kung University Hospital, College of Medicine, National Cheng Kung University, Tainan 70101, Taiwan

**Keywords:** galectin 3, adenocarcinoma of lung, neoplasm metastasis, survival

## Abstract

Background and objective: Although nodal and distant metastasis is rare in T1 lung adenocarcinoma, it is related to poor clinical prognosis. Association between galectin-3 (Gal-3) expression level, and clinical outcome of T1 lung adenocarcinoma has not been clarified. Methods: From January 2009 to December 2014, 74 patients with surgically resected T1 lung adenocarcinoma were enrolled in this retrospective cohort study. Patient outcomes were followed up until December 2019. Gal-3 expression level in primary tumors was assessed immunohistochemically and evaluated based on the staining intensity and percentage. Patient characteristics and correlation between Gal-3 expression level and clinical outcomes were reviewed. Results: Low Gal-3 expression was associated with increased metastatic events (*p* = 0.03), especially distant metastasis (*p* = 0.007), and mortality rate (*p* = 0.04). Kaplan–Meier analysis revealed that high Gal-3 expression level was associated with favorable recurrence-free survival in T1 lung adenocarcinoma (log-rank *p* = 0.048) and T1a (≤ 2 cm, American Joint Committee on Cancer (AJCC) 7th edition) lung adenocarcinoma (log-rank *p* = 0.043). Gal-3 expression along with tumor size showed a larger area under curve (AUC) than tumor size alone for predicting metastatic events (AUC = 0.747 vs. 0.681) and recurrence (AUC = 0.813 vs. 0.766) in T1a lung adenocarcinoma in the receiver-operating characteristic curve. Conclusion: Low Gal-3 expression level in primary tumors was remarkably associated with increased metastatic events and reduced recurrence-free survival in T1 lung adenocarcinoma. We suggest that Gal-3 expression level in addition to tumor size may potentially be stronger than tumor size alone in predicting metastasis in T1a lung adenocarcinoma patients.

## 1. Introduction

Lung cancer is one of the most common malignant neoplasms worldwide and its metastasis results in poor outcomes. Lung cancer cases with larger tumor size are associated with a higher metastatic rate and worse prognosis than those with small tumors [1]. However, in clinical practice, some metastases have been observed while the tumor size was in the T1 (tumor size ≤ 3 cm) category. Ye et al. [2] reported 651 consecutive patients with clinical stage IA lung cancer and found that 69 patients (10.6%) had lymph node metastasis, including 6.6% with N1 and 4% with N2 metastasis. The study based on the surveillance, epidemiology and end results (SEER) database by Yuan et al. [3] showed that 21.6% of the patients with T1 (≤ 3 cm) non-small cell lung cancer (NSCLC) and 16.6% of the T1a (≤ 2 cm) patients were diagnosed with lymph node metastasis. In addition, 7.8% T1 patients and 5.5% T1a patients were diagnosed with distant metastases. These data imply that the tumor size may not be enough to predict metastasis in T1 NSCLC.

Galectin-3 (Gal-3), a member of β-galactoside-binding lectins, is a chimeric glycoprotein encoded by a single gene, *LGALS3,* in humans, and plays multiple roles in cancer initiation, adhesion, progression, metastasis, angiogenesis, and adaptation in tumor microenvironments [4,5,6,7]. Some studies reported that Gal-3 expression level was negatively correlated with clinical outcome in lung cancer patients; however, these studies investigated different cell types and T grades of NSCLC [8,9,10,11].

Hence, the role of Gal-3 expression in T1 lung adenocarcinoma is still unclear. In this study, we examined the Gal-3 expression level and determined its association with metastasis in T1 lung adenocarcinoma.

## 2. Materials and Methods

### 2.1. Patients and Study Design

We retrospectively reviewed all patients with T1 lung adenocarcinoma diagnosed between January 1999 and December 2014 from the prospectively maintained lung cancer registry database of E-Da Hospital, a tertiary referral center in southern Taiwan. To reduce the bias that would result from intratumoral heterogeneity, only those patients who underwent tumor resection were included. Although genomic analysis is considered much better than the conventional criteria for identifying the origin of multiple lesions, [12,13] it has not been universally used in our clinical practices yet. To prevent probable arguments on distinguishing multiple primary lesions from metastatic lesions, we excluded patients who had only multiple ground-glass nodules. Patients′ survival statuses were followed up until December 2019. Any pathological or radiological evidence of involvement in lymph nodes, pleura (pleural seeding or malignant pleural effusion), and distant organs during follow-ups were considered as metastasis. The consort diagram of patient enrollment and exclusion is shown in Figure 1. This study was approved by the institutional review board of E-Da Hospital (approval number: EMRP-106-045).

### 2.2. Immunohistochemical Staining

Immunostaining for Gal-3 was performed on the fully automated Bond-Max system (Leica Microsystems, Wetzlar, Germany). Slides carrying tissue slices cut from paraffin-embedded, formalin-fixed sections were dried for 30 min at 60 °C. These slides were then covered by Bond Universal Covertiles (Leica Microsystems, Wetzlar, Germany) and placed into the Bond-Max instrument. All the subsequent steps were performed by the automated instrument according to the manufacturer’s instructions as follows: (1) deparaffinization of the tissue on the slides by rinsing with Bond Dewax Solution (Leica Microsystems, Wetzlar, Germany) at 72 °C; (2) heat-induced epitope retrieval (antigen unmasking) with Bond Epitope Retrieval Solution 2 (Leica Microsystems, Wetzlar, Germany) for 20 min at 100 °C; (3) peroxide block placement on the slides for 5 min at room temperature; (4) incubation with mouse monoclonal anti-galectin-3 antibody (Leica Biosystems, Newcastle, UK) at a dilution of 1:200 for 20 min at room temperature; (5) Bond Polymer placement (Leica Microsystems, Wetzlar, Germany) on the slides for 8 min at room temperature; (6) color development with DAB (3,3′-diaminobenzidine tetrahydrochloride) as a chromogen for 5 min at room temperature; and (7) hematoxylin counterstaining for 5 min, followed by mounting of the slides and examination by light microscopy.

### 2.3. Scoring for Gal-3 Expression Level

The sections were assessed and evaluated by two independent board-reviewed pathologists (Su-Y.C. and Liang-P.I.) who were blind to the clinical profiles and outcomes of all the patients. Gal-3 expression level was assessed by the appearance of brown particles in the cytoplasm and/or nucleus. The discordant cases were all discussed by two pathologists in order to reach the consensus results.

Immunoreactivity of Gal-3 was evaluated by a scoring system based on the intensity and percentage as described by Mathieu et al. [10]. The intensity (I) score was determined by the strength of cell staining that was graded from 0 to 2 (0: no coloration; 1: weakly positive; 2: strongly positive). The percentage (P) score was determined by the quantity of stained cells in one visual field and was graded from 0 to 2 (0: 0%; 1: 1–50%; 2: > 50%). The intensity percentage (IP) score was calculated by multiplying the intensity and percentage (I × P) values and scoring them from 0 to 4. An IP score of < 2 was considered to signify low Gal-3 expression level (Figure 2A,B), while high expression level was signified by an IP score of ≥ 2 (Figure 2C,D).

### 2.4. Data Collection and Statistical Analysis

From the institution’s prospectively maintained medical database, the demographic data, pathological stage, and clinical outcomes of all the patients were retrieved and reviewed. The pathological stage was classified based on the 7th edition of the American Joint Committee on Cancer (AJCC) staging system [14]. The tumor subtypes of each patient were quantified by a scoring system introduced by Sica et al. based on subtype grading [15,16]. Minimally invasive or lepidic predominant adenocarcinoma was graded 1; acinar or papillary predominant adenocarcinoma was graded 2; and micropapillary or solid predominant adenocarcinoma was graded 3. The subtype score was the sum of the two most prominent grades. If there was only one subtype identified, the score would be doubling the grade (e.g., a tumor with a purely acinar subtype was scored 4).

The index day was when the patients received first-line treatment (e.g., radical resection for resectable disease or systemic treatment for unresectable disease). Overall survival (OS) was defined as the interval between the index day and the patient′s death. Recurrence-free survival (RFS) was defined as the interval between the index day and the day when tumor recurrence was detected.

Pearson’s χ^2^ test was used for categorical data and Mann–Whitney U test for continuous data analysis. Kaplan–Meier analysis with log-rank test was used to compare the survival curves. Cox-regression model was used for univariable and multivariable analyses of patient survival to obtain the hazard ratio. The significance was tested two-sidedly, and a *p*-value of < 0.05 was considered statistically significant. Statistical analyses were performed using IBM SPSS Statistics for Windows, version 22 (IBM Corp., Armonk, NY, USA).

## 3. Results

Seventy-four patients were included in this study. Gal-3 expression levels were noted for the specimens obtained from all the patients. Analysis of the correlation between Gal-3 expression level in the primary tumors and the patients′ clinicopathological features revealed no remarkable association between Gal-3 expression and age, gender, T stage, tumor subtype, and lymphovascular invasion (Table 1). Low Gal-3 expression level was significantly associated with metastatic events and death (*p* = 0.03, and *p* = 0.04, respectively). Considering the site of metastasis, low Gal-3 expression level was remarkably associated with distant metastasis (*p* = 0.007). We also tested the nuclear Gal-3 expression and found that there was no significant association with metastatic events (42.9% in positive tumors vs. 53.8% in negative tumors, *p* = 0.363).

Kaplan–Meier analysis of low versus high Gal-3 expression revealed that low Gal-3 expression level was significantly correlated with poorer RFS (log-rank *p* = 0.048) and potentially with OS (log-rank *p* = 0.051) in T1 lung adenocarcinoma (Figure 3A). This correlation with RFS was also remarkable in T1a (≤ 2 cm) tumors (log-rank *p* = 0.043, Figure 3B).

In the univariable Cox regression model, Gal-3 expression was a significant prognostic factor of RFS (hazard ratio = 0.441, 95% confidence interval: 0.2–0.97, *p* = 0.042) (Table 2). In multivariable analysis, Gal-3 was also a significant prognostic factor of OS (hazard ratio = 0.193, 95% confidence interval: 0.038–0.975, *p* = 0.047).

Lastly, we investigated the value of Gal-3 expression in predicting the clinical outcomes in T1a lung adenocarcinoma. Using receiver operating characteristic (ROC) curve, adding Gal-3 expression to the tumor size (tumor size minuses Gal-3 expression level) showed a larger area under the curve (AUC) than tumor size alone in both predicting events of metastasis (AUC = 0.747 vs. 0.681, Figure 4A) and recurrence (AUC = 0.813 vs. 0.766, Figure 4B).

## 4. Discussion

Gal-3 is a multifunctional signaling molecule involved in tumor development and cell-cell interactions in the tumor microenvironment [17]. Although many studies have indicated that Gal-3 promotes cancer progression, there have also been several evidences showing its dual activities. Based on its location, Gal-3 supports or suppresses tumor progression in prostate cancer [18,19]. Via different pathways, Gal-3 plays distinct roles in melanoma metastasis and leads to reverse correlations with disease prognosis [20,21].

In lung cancer, Gal-3 was previously shown to promote cancer aggressiveness in vitro [7]. However, the association between Gal-3 expression and patient survival is still disputable. Szöke et al. [8] studied 94 lung cancer patients and found that cytoplasmic Gal-3 expression level indicated poor prognosis in patients with stage II NSCLC. Two independent studies by Puglisi et al. [9] and Mathieu et al. [10] showed that nuclear Gal-3 expression level, and not cytoplasmic expression, was significantly correlated with poor survival in NSCLC patients. In contrast, Jeong et al. [22] suggested that a high Gal-3 expression level was not associated with poor survival in NSCLC patients. There are possible explanations for this phenomenon. First, Gal-3 expression is totally different in adenocarcinoma and squamous cell carcinoma [10,23]. Gal-3 expression in 2D monolayer cultures is also distinct from that in 3D cultures or in vivo assays [24]. Hence, studies including multiple cell types or in vitro experiments may present different results. Second, intratumoral heterogeneity may result in misleading information on the real Gal-3 expression level, especially if the tissue source is minimal, such as that obtained by needle biopsy. Our experience showed that Gal-3 expression level could be heterogeneous in one tumor (see Appendix A). Third, Gal-3 may play multiple roles during lung cancer progression, leading to distinct expressions at each stage. Unlike the in vitro studies that were generally designed around Gal-3 knockout models, we noted that all primary tumors in our cohorts expressed Gal-3 in the cytoplasm but at different levels. This suggested us to investigate Gal-3 expression level in a manner of “high or low” rather than “yes or no”.

Literature review indicates that low Gal-3 expression level may be probably beneficial for cancer metastasis in some conditions. Gal-3 down-regulation in primary tumors could promote cancer progression and metastasis in human breast cancers and melanomas [21,25,26]. Decreased ability for DNA repair, susceptibility to DNA damage, and uncontrolled proliferation are important hallmarks of cancer development. Kosacka et al. [27] examined the expression levels of Gal-3 and cyclin D1 in 47 NSCLC tissues and reported a negative correlation with lung adenocarcinoma, wherein they reported higher cyclin D1 expression in low Gal-3 tumors. Carvalho et al. [28] exposed different DNA damaging agents to human cells to investigate the role of Gal-3 in DNA damage repair pathways. They found that, in the absence of Gal-3, there was a delayed activation of DNA repair response and decrease in the gap 2-to-mitosis (G2/M) cell cycle checkpoint arrest. These studies suggested that low Gal-3 expression may be associated with a decreased ability of DNA repair and increased proliferation in lung cancer cells, which could promote tumor aggressiveness.

In addition to deregulated cell proliferation, immune escape is vital for cancer metastasis. Gal-3 has been reported to act as an alarmin and a novel chemo-attractant for monocytes, macrophages and neutrophils, thereby facilitating the release of proinflammatory cytokines [29,30,31,32,33]. Reduced Gal-3 expression was shown to decrease inflammatory responses [34,35]. In the tumor microenvironment, regulatory T cells (Tregs) and M2 macrophages are involved in the process of immune escape. Gal-3 was found to inhibit differentiation of naive T cells into Tregs and polarize macrophages toward an M1 phenotype [36]. Putting this information together, a higher Gal-3 expression probably induces a stronger inflammatory response than lower Gal-3 expression, which may not be good for a small tumor to disseminate from the primary site.

In our study, we enrolled patients with resected T1 lung adenocarcinoma. This criterion facilitated complete review of whole primary tumors and prevented the bias from intratumoral heterogeneity and different cell types. All patients have been followed for at least 5 years to avoid bias from occult or delayed metastasis, which may be seen in a cross-sectional study. This approach provided solid evidence of the relationship between Gal-3 expression level and patient outcomes. To the best of our knowledge, our study is the first to report on Gal-3 expression level in T1 lung adenocarcinoma and its association with metastatic events and patient survival. We found that low Gal-3 expression level in the primary tumor was significantly associated with increased metastasis events and reduced RFS. We also noted that Gal-3 expression level could be of value in predicting metastasis in T1a lung adenocarcinoma patients.

There were some limitations to our study. First, although we tried our best to search for all candidates from the medical record database, there could still have been some sample losses because of the retrospective design. The small sample size limited the multivariate analysis, leading to insignificant results. Next, we selected resected lung adenocarcinoma to decline bias from intratumoral heterogeneity but we also increased the selection bias. We also excluded patients who only had multiple lung ground-glass nodules because we were unable to confirm the etiology. The data from our study should be validated in patients who only have lung-to-lung metastasis and in large-scale retrospective cohorts and prospective cohorts to achieve precise evaluations for clinical utility [37].

In conclusion, we found that low Gal-3 expression level in primary tumors was remarkably associated with increased metastasis events and reduced RFS in T1 lung adenocarcinoma. Our study findings implicate that using Gal-3 expression levels in addition to tumor size would be more beneficial than using tumor size alone for the prediction of metastatic events in T1a lung adenocarcinoma.

## Figures and Tables

**Figure 1 jcm-09-01990-f001:**
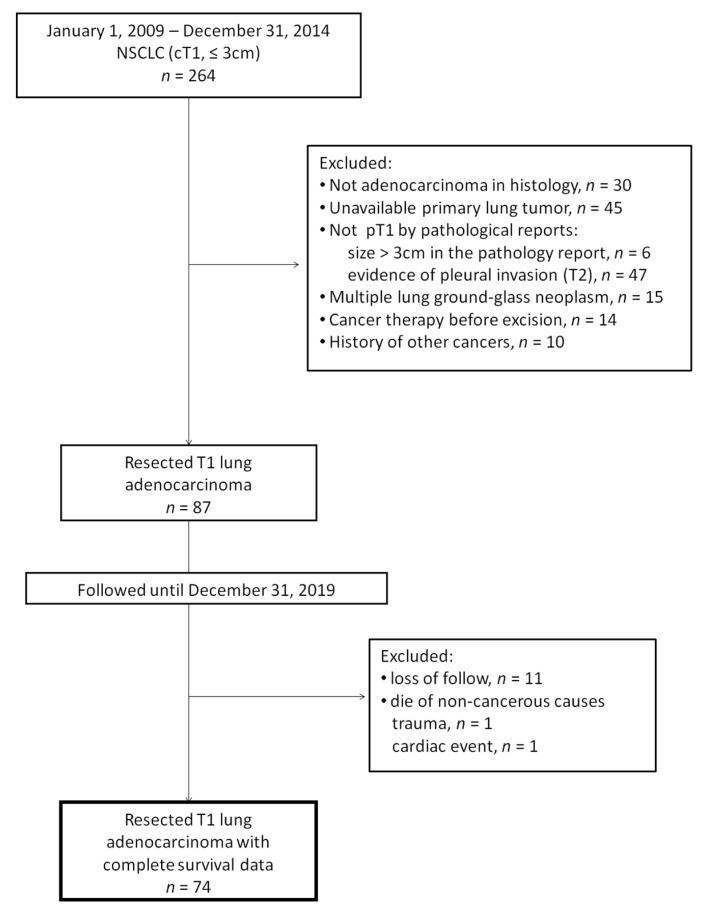
Consort diagram. NSCLC: non-small cell lung cancer; cT1: clinical stage T1; pT1: pathological stage T1.

**Figure 2 jcm-09-01990-f002:**
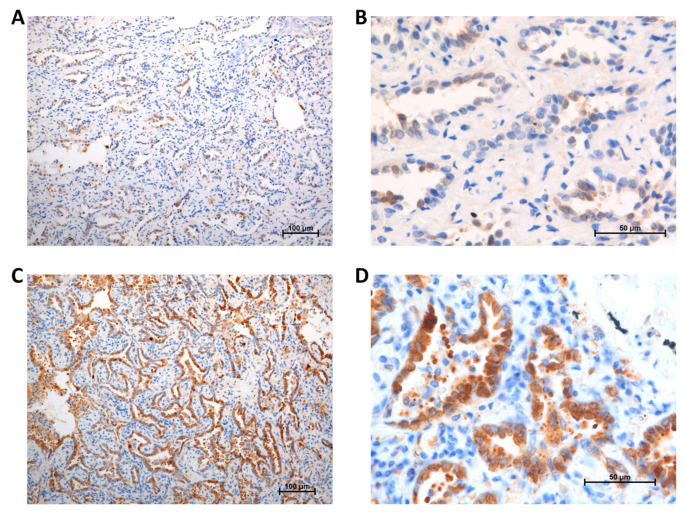
Galectin-3 expression: (**A**) low, 100 ×; (**B**) low, 400 ×; (**C**) high, 100 ×; (**D**) high, 400 ×.

**Figure 3 jcm-09-01990-f003:**
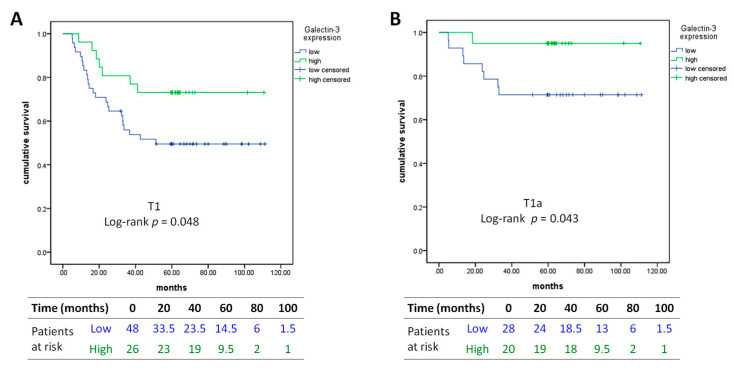
Kaplan–Meier curves of recurrence-free survival in (**A**) T1 lung adenocarcinoma and in (**B**) T1a lung adenocarcinoma. Low: intensity percentage (IP) score < 2; high: IP score ≥ 2.

**Figure 4 jcm-09-01990-f004:**
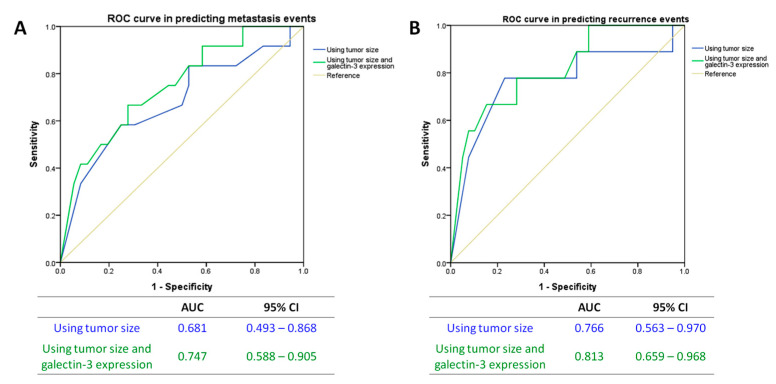
ROC curves in the prediction of (**A**) metastasis events and (**B**) recurrence events. ROC: receiver operating characteristic; AUC: area under curve; CI: confidence interval.

**Table 1 jcm-09-01990-t001:** Patient characteristics by galectin-3 expression.

Galectin-3 Expression	Low *n* = 48	High *n* = 26	*p*-Value
Age (Year)	63 (55–70)	62.5 (57.5–69.5)	0.738
Gender, Female	26 (54.2)	20 (76.9)	0.079
T Stage *			0.132
T1a (≤ 2 cm)	28 (58.3)	20 (76.9)	
T1b (2–3 cm)	20 (41.7)	6 (23.1)	
Subtype Score	4 (3–7.5)	4 (3–4)	0.924
Lymphovascular Invasion	9 (18.8)	2 (7.7)	0.309
**Outcomes**			
**Metastasis, All Kinds**	28 (58.3)	8 (30.8)	0.03
**Metastasis, Nodal**	15 (31.3)	5 (19.2)	0.411
**Metastasis, Distant**	25 (52.1)	5 (19.2)	0.007
**Recurrence**	24 (50)	7 (26.7)	0.084
**Mortality**	14 (29.2)	2 (7.7)	0.04

Data are presented as number (%) or median (interquartile range). * The T stage is according to the 7th edition of American Joint Committee on Cancer (AJCC) staging system.

**Table 2 jcm-09-01990-t002:** Univariable and multivariable analysis for overall survival and recurrence-free survival.

**Univariable Analysis**	**Overall Survival**			**Recurrence-Free Survival**		
**Variables**	**Hazard Ratio**	**95% Confidence Interval**	***p*-Value**	**Hazard Ratio**	**95% Confidence Interval**	***p*-Value**
Age (Year)	1.036	0.986–1.089	0.159	1.002	0.970–1.035	0.91
Gender (Male vs. Female)	8.473	2.412–29.765	0.001	3.201	1.562–6.562	0.001
T Stage (T1b vs. T1a) *	6.229	2.123–18.275	0.001	7.822	3.542–17.274	<0.0001
Subtype Score	1.897	1.234–2.917	0.004	1.478	1.063–2.054	0.02
Lymphovascular Invasion (Yes vs. No)	17.162	5.778–50.972	<0.0001	3.476	1.533–7.884	0.003
Galectin-3 (High vs. Low)	0.254	0.057–1.122	0.071	0.441	0.2–0.97	0.042
**Multivariable Analysis**	**Overall Survival**			**Recurrence-Free Survival**		
**Variables**	**Hazard Ratio**	**95% Confidence Interval**	***p*-Value**	**Hazard Ratio**	**95% Confidence Interval**	***p*-Value**
Gender (Male vs. Female)	2.563	0.608–10.798	0.200	2.346	1.034–5.324	0.041
T Stage (T1b vs. T1a) *	2.641	0.719–9.704	0.143	6.884	2.735–17.324	<0.0001
Subtype Score	1.845	1.052–3.233	0.032	0.888	0.581–1.357	0.583
Lymphovascular Invasion (Yes vs. No)	23.934	4.882–117.344	<0.0001	1.493	0.608–3.664	0.382
Galectin-3 (High vs. Low)	0.193	0.038–0.975	0.047	0.794	0.330–1.908	0.606

* The T stage is according to the 7th edition of the American Joint Committee on Cancer (AJCC) staging system.

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
