# Peer review of "Low Galectin-3 Expression Level in Primary Tumors Is Associated with Metastasis in T1 Lung Adenocarcinoma"

_jcm, 2020, doi:10.3390/jcm9061990_

Round 1
Reviewer 1 Report
This manuscript documents a retrospective correlation of low galectin-3 expression levels and development of metastases as well as reduced recurrence free survival in patients presenting with T1 adenocarcinoma of the lung. It is well written. The Discussion could be shortened. As the authors themselves note, the sample size is small and this impacted the multivariate analysis. This study does not confirm low galactin-3 as prognostic biomarker however. Clinical validity and clinical utility must still be proven. An evaluation for clinical utility could be performed through a prospective/retrospsective study with archived specimens (see: Hayes DF. Biomarker validation and Testing. Molecular Oncology 9 (2015) 960-966). You could mention this in the discussion.
Two small grammatical/phrasesology comments:
1) Lines 53-54: rewrite as "these studies investigated different cell types and T grades of NSCLC."
2) Page 9 paragraph 4: facilitated complete review of whole...
Reviewer 2 Report
Major comment)
The authors examined Galectin-3 expression level in T1 lung adenocarcinoma, and found low Galectin-3 level and correlation with prognosis and metastasis. The paper is well written and arranged.
Minor comment)
In Figure 2, it presents 40x and 100x pictures, however, the size does not change much. You may use a slightly different magnification, or either one.
Reviewer 3 Report
First of all, I would like to congratulate the authors for this interesting study where they revealed new insights about Galectin-3 and its association with clinical outcomes. In this article, the authors revealed the association between Gal-3 expression levels in primary tumours with metastatic events and PFS in T1 lung adenocarcinoma.
Major issues:
1.About scoring for Gal-3 Expression Level, the authors may include how you resolve discordant IHC evaluations between the two independent board-reviewed pathologists (Page 4, Line 92-94).
2.In Results sections, authors may consider including the table of multivariable analysis like as you are done with the univariable analysis (Page 6, Line 149-150).
3.In Figure 3, authors might indicate in the legend of Kaplan Meyer the following statement ‘’ low (score <2) and high (score >2) ‘’
4.In the Discussion section, authors mentioned that their experience showed ‘’ The intratumoral heterogeneity may result in misleading information on the real Gal-3 expression level, especially if the tissue source is minimal, such as that obtained by needle biopsy so Gal-3 expression level could be heterogeneous in one tumour ‘’. To avoid this heterogeneity, have you considered the use of the liquid biopsy or other methodology?
5. It would be recommended the validation of Gal-3 levels results in a prospective validation cohort or external alternative cohort.
6. Authors may be considered to validate their results with other methodology, as measuring gene expression through RT-qPCR.
Round 2
Reviewer 3 Report
Different suggestions have been modified.
Other studies should be performed in the future to confirm these results.